# The greenhouse gas impacts of converting food production in England and Wales to organic methods

Laurence G. Smith [1,2], Guy J.D. Kirk [1*], Philip J. Jones [3] & Adrian G. Williams[1]

Agriculture is a major contributor to global greenhouse gas (GHG) emissions and must feature in efforts to reduce emissions. Organic farming might contribute to this through decreased use of farm inputs and increased soil carbon sequestration, but it might also exacerbate emissions through greater food production elsewhere to make up for lower organic yields. To date there has been no rigorous assessment of this potential at national scales. Here we assess the consequences for net GHG emissions of a 100% shift to organic food production in England and Wales using life-cycle assessment. We predict major shortfalls in production of most agricultural products against a conventional baseline. Direct GHG emissions are reduced with organic farming, but when increased overseas land use to compensate for shortfalls in domestic supply are factored in, net emissions are greater. Enhanced soil carbon sequestration could offset only a small part of the higher overseas emissions.

[1] School of Water, Energy & Environment, Cranfield University, Cranfield MK43 0AL, UK. [2] School of Agriculture, Food and Environment, Royal Agricultural University, Cirencester GL7 6JS, UK. [3] School of Agriculture, Policy and Development, University of Reading, PO Box 237Reading RG6 6AR, UK. *email: g.kirk@cranfield.ac.uk

Organic farming is often suggested as a solution to the negative environmental effects of current food production[1]. Reduced farm inputs and more soil carbon sequestration may alter local GHG budgets favourably. But this must be set against the need for increased production and associated land conversion elsewhere as a result of lower crop and livestock yields under organic methods.

Past studies of the potential of organic farming to mitigate GHG emissions have produced mixed results[2]. For example, Williams et al.[3] found that most organic cropping systems in England generate similar or greater GHG emissions per tonne of crop compared with conventional systems, with lower yields and increased rates of nitrate leaching offsetting the lower use of inputs. Conversely, a Swiss study, which considered entire crop rotations and less-intensive modes of production than Williams et al.[3], found much lower GHG emissions per tonne of organic crop[4]. Studies comparing organic and non-organic livestock production have also yielded mixed results. In dairy production, reduced use of inputs per tonne of milk under organic management is offset by lower milk yields and lower feed conversion ratios[3,5]. Whereas organic beef and sheep production systems can have greater environmental efficiencies as a result of the replacement of manufactured nitrogen (N) fertiliser with biologically-fixed N from forage legumes[6–8]. In organic poultry production, reduced productivities and low feed conversion ratios considerably reduce environmental efficiencies[9–11]. Similarly, organic pig production tends to have lower environmental efficiencies per tonne of product due to lower stocking densities and less output per hectare[12,13]. Even where environmental efficiency per hectare is improved, organic systems require more land per tonne of product as a result of lower yields: Williams et al.[3] found additional land requirements of from 65 to 200%.

The most recent attempt to quantify the GHG mitigation potential of organic farming at a national scale was made by Audsley et al.[14], who used a life-cycle assessment model (LCA) to compare UK organic and conventional data on commodity production, processing, distribution, retail and trade. A 'baseline' LCA based assessment, reflecting actual consumption patterns, was compared with a range of scenarios, one of which was a transition to 100% organic production. This built on a study by Jones and Crane[15] in which the production impacts of a 100% conversion to organic agriculture in England and Wales were estimated using data on organic yields, crop areas and livestock numbers from the Farm Business Survey. The results indicated that a switch to organic production in the UK could result in a GHG emission reduction of about 8% in terms of UK production. However, the emissions associated with the additional land use changes overseas required to meet UK supply shortfalls were not considered.

In an earlier study[16], we developed a model to estimate potential maximum food production from all agriculture—crops and livestock—in England and Wales under organic management. In this paper we extend this analysis to estimate effects on national GHG balances. We assess the impacts of conversion of all agriculture to organic farming using the Agri-LCA models developed by Williams et al.[3] to estimate GHG emissions from individual agricultural systems. This includes carbon dioxide ($CO_2$) emissions from fossil energy use in farm operations and in the production and transport of farm inputs and outputs, as well as emissions of methane ($CH_4$) and nitrous oxide ($N_2O$) as functions of soil conditions, nutrient management and livestock variables (Methods). We improved on the Audsley et al.[14] assessment by also accounting for, first, limits to organic production imposed by the supply of livestock feed, rotational constraints and available N, second, the GHG impact of overseas land use changes associated with increased food-imports, and third,

the GHG offset potential of soil carbon (C) sequestration under organic production. We also estimate uncertainties in our calculations using Monte Carlo analyses. In doing so we provide the most comprehensive national-scale assessment to-date of the potential land use, production and GHG impacts of up-scaling organic agriculture.

## Results

**Predicted food production.** We predict a drop in total food production expressed as metabolisable energy (ME) by of the order of 40% compared to the conventional farming baseline (Fig. 1, Supplementary Table 1). Human edible protein outputs decreases by a similar proportion (Supplementary Table 2). The decrease is due to smaller crop yields per unit of land area under organic management, and the need to introduce fertility-building grass leys with nitrogen-fixing legumes within crop rotations. The latter requirement is a farming system-level effect that is not captured in crop-level comparisons[16–18].

Figure 1 also shows large shifts in the combination of crops grown and numbers of animals reared. Increased diversity of crop

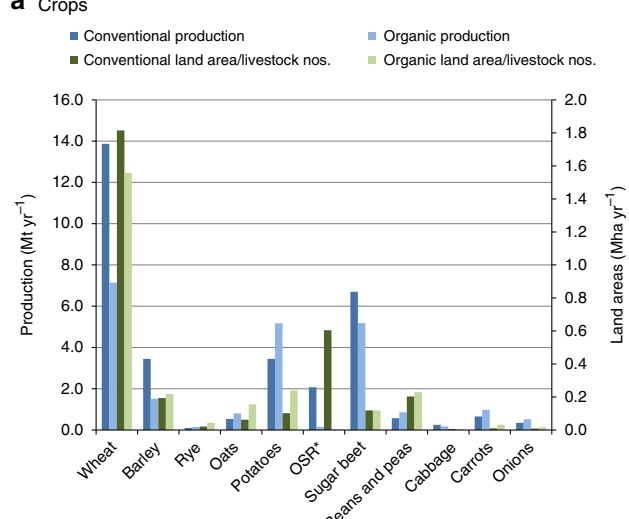

**a** Crops

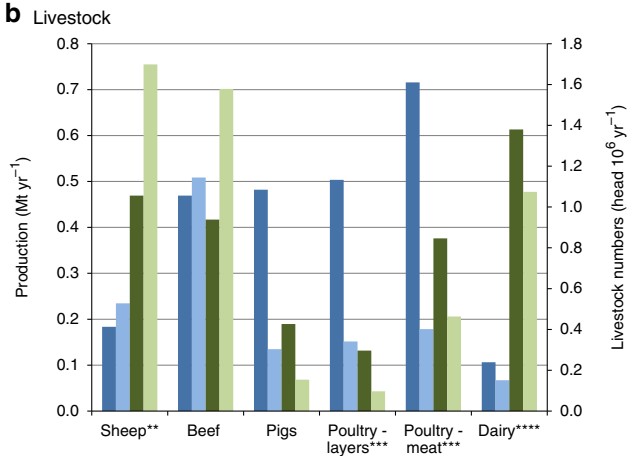

**b** Livestock

**Fig. 1** Projected food production under conventional and organic farming methods. **a** Crop production and areas. *oilseed rape. **b** Livestock production and numbers. **sheep numbers × 10, ***poultry numbers × 100, ****milk production in Mt × $10^5$. Conversion to 100% organic methods caused decreases in wheat, barley, oilseed rape, pigs, eggs, poultry meat and milk, and an overall decrease to 64% of the conventional baseline. Data of Smith et al.[16]. Source Data are provided as a Source Data file

rotations under organic management means total vegetable production is maintained[16]. Edible protein production increases in arable areas, particularly in the east and north east of England, through increases in ruminant livestock and legume production[16]. Production of organic oilseed rape (OSR) decreases substantially, primarily because of a much smaller cultivated area due to the relatively low yield of organic OSR compared to both conventional OSR and organic alternatives. The increase in legume and potato production is a result of an increase in the cultivated area: legumes for biological N fixation and potatoes both for weed control and because of their high ME yield. The area would have increased further had the constraint on maximum production in the model not been reached, which we set at 150% of current supply to reflect limits on consumer demand[19,20]. Total sugar beet production decreased, but, due to its high ME yield, it reached its upper local limit in parts of eastern England, which we imposed to restrict expansion away from major processing centres[16]. For most crops, the projected decreases in output are considerably greater than might be expected solely from the displacement of crops with leys in organic rotations. The production of minor cereals, such as oats and rye, increases, but this is not sufficient to offset the losses of wheat and barley.

Numbers of grazing livestock (sheep and beef cattle less dairy) increase, because of the increase in feed availability from leys. But the volume of meat produced did not increase in proportion, as a result of lower carcass weights and longer finishing times under organic management. Numbers of monogastric livestock (pigs and poultry) and associated meat production fell sharply as a result of lower stocking rates and availability of concentrated feed. Dairy cattle numbers and milk production decrease due to greater reliance on concentrated feeds than grazing livestock and hence greater sensitivity to N availability, cropping area and cereal yields.

**GHG emissions per unit production**. Figure 2a shows estimated GHG emissions per unit of production for individual crops. The lower GHG emissions under organic cropping are largely due to replacement of N fertiliser with biological N fixation in leys, resulting in less $CO_2$ and $N_2O$ from fertiliser manufacture and less $N_2O$ per unit of production[3,4,21]. We concentrate on N in our analysis, and not on other plant nutrients, because N is required in the greatest quantities and its inputs and outputs are the most sensitive to differences between conventional and organic systems. However, balances of P, K and other nutrients must also be maintained, and we therefore account for the GHGs associated with extracting and applying the P and K minerals commonly used in organic systems to maintain balances.

Emissions per unit production are greater for some organic crops, such as field beans, due to increased N leaching and nitrification-denitrification losses, because more must be grown on heavy wet soils. However, a large proportion of field beans grown would have to be exported because of low rates of domestic consumption, and we allow for this in the model with a maximum limit on production, as for potatoes. Oats and spring barley, which require less manufactured N fertiliser than other cereals, have greater GHG emissions per unit production under organic management because yields are smaller. Lower marketable yields in organic potato cropping also lead to greater emissions per unit of product[22]. Emissions are also greater for organic crops requiring higher fossil fuel input in their cultivation, such as organic carrots requiring flame weeding.

Figure 2b shows emissions per unit of production for individual livestock types. Organic pig production results in lower GHG emissions per unit of production because outdoor organic systems use less fossil energy in housing and there are no $CH_4$ emissions from slurry storage; however, $N_2O$ emissions increase as a result of greater leaching and denitrification from organic manures. In common with previous studies, we find that poultry meat and egg production generates greater emissions under organic management due to poorer feed conversion ratios, longer rearing times, higher mortality rates and greater leaching losses compared to conventional free range and fully housed systems[9,10]. Organic dairy, beef and sheep production results in lower total GHG emissions per unit of production, as a result of the increased efficiency of forage production under organic management, although greater forage intake increases the total $CH_4$ contribution.

**National GHG emissions**. Figure 3 gives the aggregated national emissions. It shows that the direct emissions associated with organic crop (Fig. 3a) and livestock (Fig. 3c) production are smaller for organic farming compared with conventional: by 20% for crops, 4% for livestock and 6% overall. This is a slightly lower estimate of the effect of conversion to organic farming than in Audsley et al.'s study[14]. The decrease occurs despite an increase in transport emissions, illustrating the relatively small contribution that transport makes to agriculture's total GHG budget[23].

However, the picture is very different when we allow for, first, $CO_2$ emissions from land use change overseas to make up for shortfalls in home production under organic methods, and second, enhanced soil C sequestration under organic methods at home and overseas, as shown in Fig. 3b, and 3d for different ways of making these allowances. The next two sections give our rationale for how we have done this.

**Soil carbon sequestration**. Carbon sequestration rates are expected to be greater under organic farming because of greater use of manures and slurry linked to more integrated management of livestock and crops, and longer crop rotations with leys involving forage legumes[24]. Although in conventional systems there is generally a greater separation of livestock from crops, farmyard manures will mostly be applied to land somewhere, so the net transfer of C from the atmosphere to land would be about the same[25,26]. On the other hand, excessive manure applications in livestock-dense areas under conventional management leads to over-fertilisation and suboptimal C sequestration[27]. Although we found livestock production decreased under organic management, total livestock numbers were not much different and there was a substantial shift to grazing animals with 61% more sheep and 14% more cattle (beef plus dairy; Fig. 1). We estimate there would be approximately 12% more farmyard manure as a result (Supplementary Table 3).

We estimate potential C sequestration under organic management using rates of change in soil C derived from the National Soil Inventory of England and Wales for different land use classes by Kirk and Bellamy[28], and assuming the change from conventional to organic farming was equivalent to a change from continuous arable cropping to rotational grass (Methods). This gives sequestration rates of 0.28 Mg C ha$^{-1}$ yr$^{-1}$ for arable land converted to rotational grass, or, after adjusting for the proportion of arable to arable plus rotational grass across England and Wales, 0.18 Mg C ha$^{-1}$ yr$^{-1}$. We used this as the upper rate in the calculations for Fig. 3. For comparison, in a literature review of experiments comparing conventional and organic farming, Gattinger et al.[24] found sequestration rates between 0.07 and 0.45 Mg C ha$^{-1}$ yr$^{-1}$. However, most of these comparisons involved very high rates of external organic matter inputs to the organic systems, up to 4 times those under conventional farming[26]. Given that we found only 12% more farmyard manure

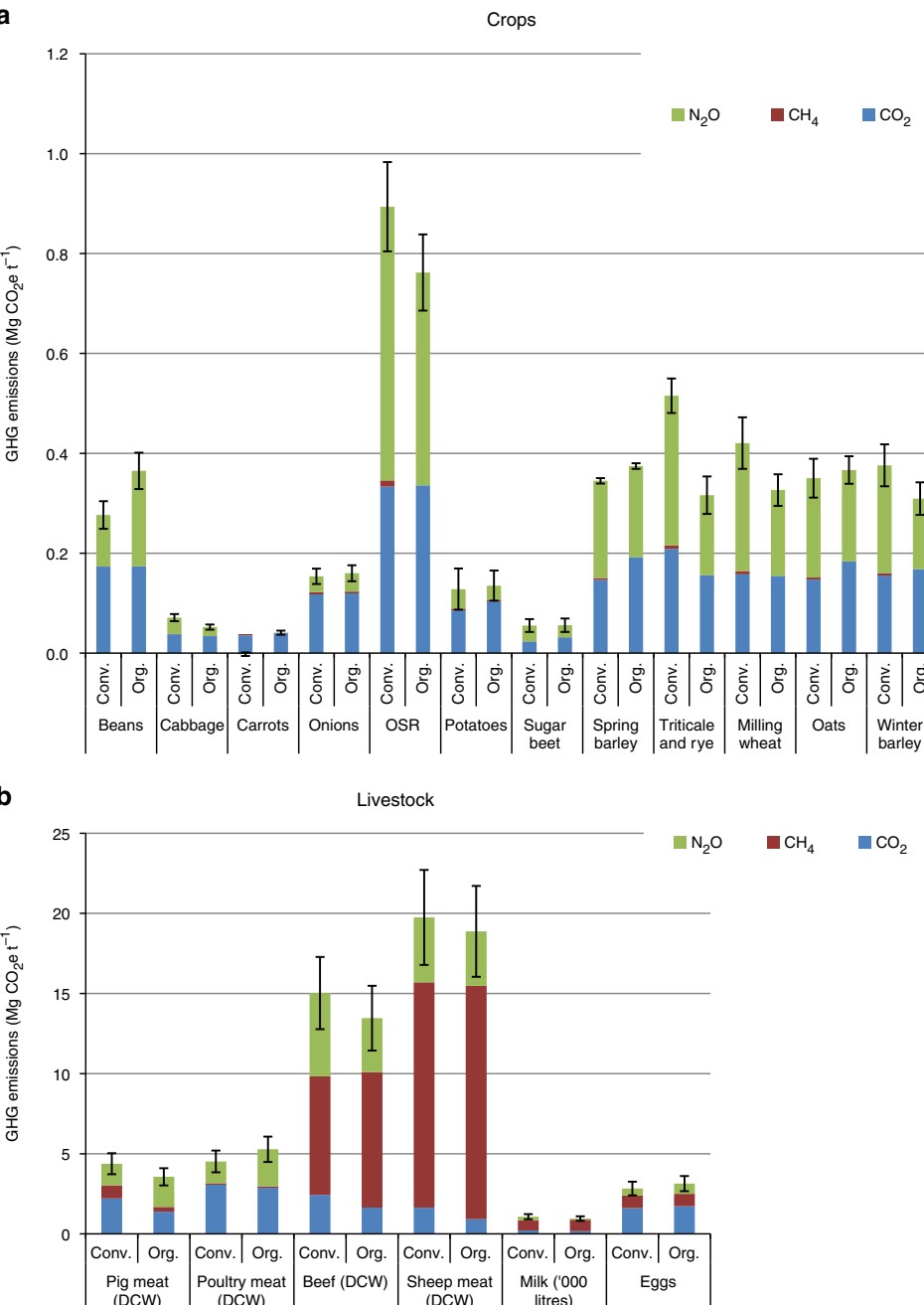

**Fig. 2** GHG emissions per unit production under conventional and organic farming methods. **a** Crops. **b** Livestock, including emissions in feed production. $N_2O$ = nitrous oxide, $CH_4$ = methane, $CO_2$ = carbon dioxide. Production is expressed in tonnes (t) of total metabolised energy. Data are means ±1 standard deviations from the uncertainty analysis (Methods). Emissions due to land use change overseas to compensate for shortfalls in home production, and enhanced soil carbon sequestration under organic methods, are not allowed for. Organic dairy, beef and sheep production have lower total GHG emissions per tonne of product, although greater forage intake increases $CH_4$ emission. Less N fertiliser use in organic farming gives $N_2O$ and fossil energy use savings per tonne of product. Exceptions are crops receiving less N fertiliser in conventional farming (beans, oats), organic crops requiring flame weeding (carrots) and organic vegetable crops with lower marketable yields (potatoes, onions). Source Data are provided as a Source Data file

under organic farming, Gattinger et al.'s higher estimates are unrealistic. We therefore use Gattinger et al.'s[24] lower value as the moderate rate in Fig. 3.

It should be noted that the bulk of any C sequestration will be limited to the first decade or two following conversion, because any given soil has a finite capacity to accumulate C depending on its characteristics and local environmental conditions[25,29,30]. A new steady-state soil C content will be reached after a few decades when rates of decomposition in the soil at the higher C content match the increased rates of C inputs.

**Overseas land conversion**. We estimate that the land area needed to make up for shortfalls in domestic production is nearly five times the current overseas land area used for food for England and Wales (Fig. 4). Total agricultural land-use is therefore 1.5 times greater than the conventional baseline (combining domestic and overseas land). This is considerably greater than the 16–33% increase in land requirements projected in a recent study of global conversion to organic farming[31]. The difference reflects the high conventional crop yields and livestock productivity in the UK compared with countries using less intensive, lower-yielding

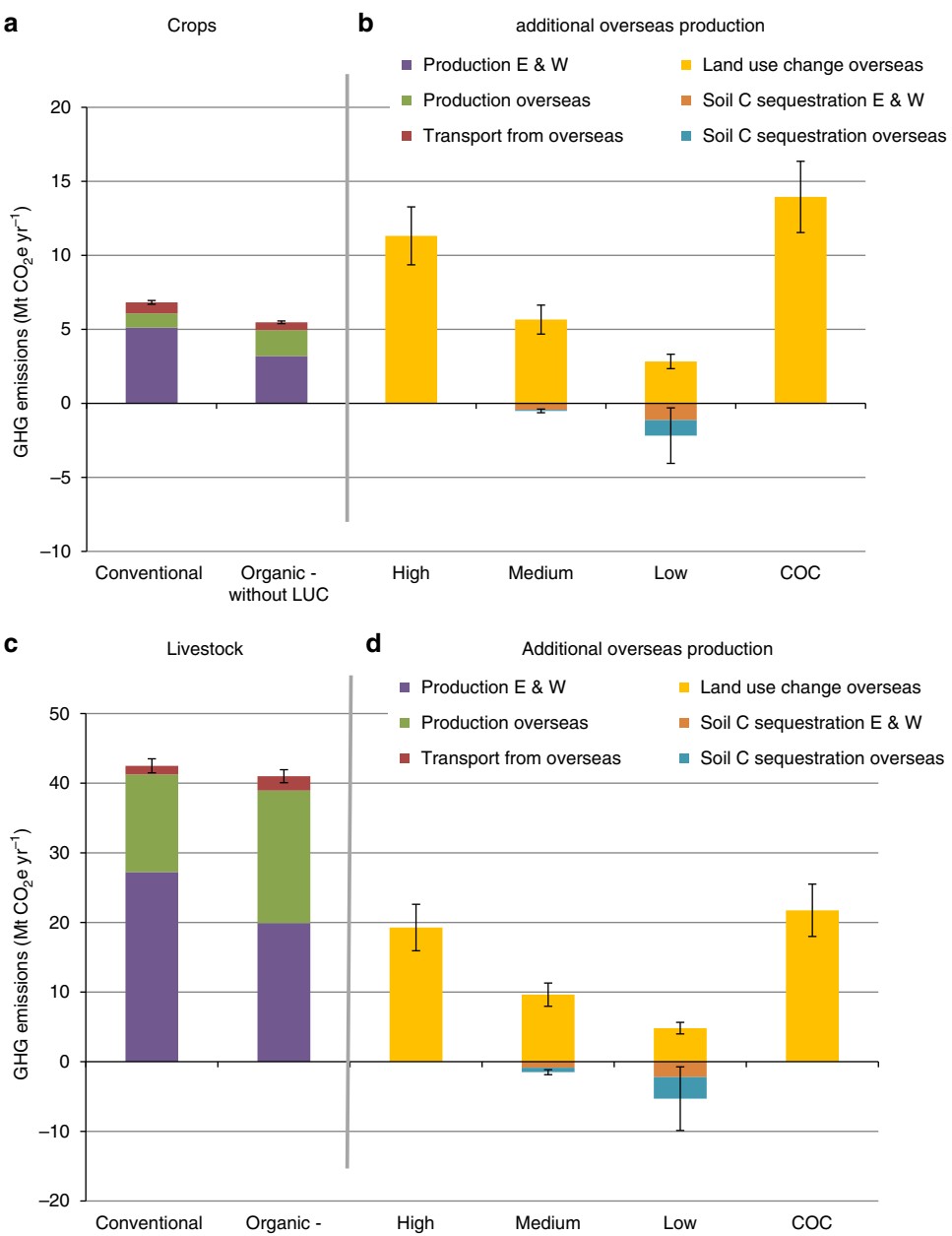

**Fig. 3** Total GHG emissions from food production for England and Wales (E & W) under conventional and organic farming methods. **a** For food crops for human consumption both from home and overseas production. **b** Additional net emissions due to soil C sequestration (CS) and overseas land use changes (LUC) to compensate for shortfalls in home production: High = all LUC by conversion from grassland, no CS; Medium = 50% of LUC by conversion from grassland, moderate CS; Low = 25% of LUC by conversion from grassland, high CS; COC= carbon opportunity cost of Searchinger et al.[35] (Methods). **c** For livestock both from home and overseas production, including emissions during production of crops fed to livestock. **d** Additional net emissions due to CS and overseas LUC to compensate for shortfalls in livestock production: High, Medium, Low as for Crops. Note LUC losses and CS gains both only apply over the first few decades following conversion, however a flat rate is applied here. Data are means ±1 standard deviation from the uncertainty analysis (Methods). Source Data are provided as a Source Data file

farming, and the correspondingly greater production penalties in conversion to organic methods[32].

The consequences for net GHG emissions will depend on the nature of the land use change. If it entails conversion of existing natural or semi-natural vegetation or pasture to crops, the cost will be greater than for increased production from existing arable land, which will have already lost C compared with its original natural state, and which might be expected to sequester some C from the atmosphere under organic management. The emissions associated with land use changes will apply over a similar period

to the potential gains from enhanced soil C sequestration (i.e., a few decades). We compare three ways of assessing this and associated soil C sequestration: first, if all the additional production is on land formerly under grass, with no associated C sequestration; second, if half the additional production is on land formerly under grass, with a low rate of C sequestration; and third, if a quarter of the additional production is on land formerly under grass, with a high rate of C sequestration (Methods).

In addition, there is the opportunity cost of the amount of C that could be sequestered if the land were instead used to

maximise its C storage potential, for example by converting it to productive forest. This aspect is considered by Searchinger et al.[35], who define a 'Carbon Opportunity Cost' (COC) as the amount of C that could be sequestered annually per kg of agricultural commodity if the land were instead used to regenerate forest. We also calculated this (Methods).

The results (Fig. 3b, d and Table 1) show that the net effects are sensitive to both the LUC scenario and the degree of soil C sequestration. If all the LUC is by conversion of grassland with no C sequestration (the High scenario), net emissions increase by 56% over the conventional baseline. Whereas, if only 25% of the LUC is from grassland, with a high rate of C sequestration (the Low scenario), net emissions are comparable to those in the conventional baseline. With 50% LUC from grassland, and a moderate rate of C sequestration (the Medium scenario), the net increase is 21%. However, if the COC is added in, the net GHG costs of organic production are much worse. For the Medium LUC and C sequestration scenario, adding in the COC ($35.7 \pm 6.6$ Mt $CO_2$e yr$^{-1}$) gives a net increase in emissions over the conventional baseline of 1.7 times.

## Discussion

The results show that widespread adoption of organic farming practices would lead to net increases in GHG emissions as a result of lower crop and livestock yields and hence the need for additional production and associated land use changes overseas. It is

not obvious how additional overseas land could be found, without expanding the existing area of tilled land by ploughing up grassland. The global demand for food is expected to increase by 59–98% by 2050[34]. Given that land resources are finite, this implies more competition for land, and more-intensive food production per unit land area, whereas current organic systems are inherently less intensive.

There are undoubted local environmental benefits to organic farming practices, including soil C storage, reduced exposure to pesticides and improved biodiversity. However, these potential benefits need to be set against the requirement for greater production elsewhere. As well as increased GHG emissions from compensatory changes in land use to make up for production shortfalls, there are substantial opportunity costs from reduced availability of land for other purposes, such as greater C storage under natural vegetation[35]. Further, although organic systems may favour increased local biodiversity, habitat fragmentation under low-yielding organic systems may mean global species diversity is in fact greater under land-sparing, high-yielding systems[36,37].

Could yields under organic management be improved to reduce land requirements? Improvements in organic rotation design and more effective and reliable supplies of N from biological fixation are possibilities[38,39]. However, these improvements are probably marginal, given the fundamental requirement for more leys in rotations under organic management. Given the much larger contribution of livestock farming to GHG emissions, a greater impact could be gained from reduced meat consumption. Less livestock farming could release land for crops for human consumption and for other purposes such as C storage[40]. However, against this, global trends are towards greater per capita and total meat consumption[33]. Also livestock can play important roles in local nutrient cycling and the provision of ecosystem services[41,42].

In summary, our assessment of the impacts of a 100% conversion to organic farming in England and Wales has revealed that, whilst improvements in resource use efficiency could be obtained, reduced outputs would mean that more imports would be required to maintain food supplies. This major expansion in agricultural cultivation overseas to make up for domestic supply shortfalls would lead to increased GHG emissions from the associated land use changes. Ultimately it is unlikely that there exists any single optimal approach to achieving environmentally sustainable food production. Therefore, context-specific evaluations are required to reveal the extent to which organic systems can contribute, alongside other approaches, to multi-objective and internationally binding sustainability targets.

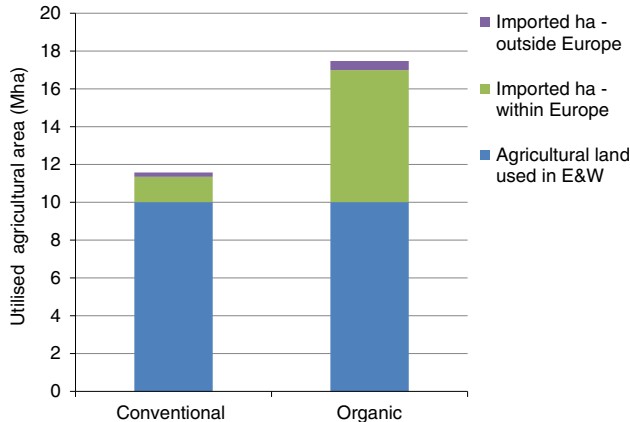

**Fig. 4** Overseas land area needed for imported food. The area required to offset shortfalls in domestic production under organic methods is over five times that under conventional methods, largely due to imports of oilseeds, pork, poultry meat, eggs and milk. Note only the products listed in Fig. 1 are included; products that are not produced in the UK on a large scale (such as maize, rice, tea, coffee and sugar cane) are excluded (Methods). Source Data are provided as a Source Data file

## Methods

**The OLUM.** The OLUM (Optimal Land Use Model)[16] is a linear programming (LP) model that includes a suite of organic farming activities that take place in nine Robust Farm Types: specialist cropping, mixed arable and livestock, specialist

**Table 1 Total GHG emissions from crop and livestock production under conventional and organic production allowing for High, Medium and Low levels of overseas LUC and soil C sequestration as in Fig. 3**

|  | Conventional | Organic | | |
|---|---|---|---|---|
|  |  | High | Medium | Low |
| Emissions (Mt $CO_2$e yr$^{-1}$) | $49.3 \pm 2.1$ | $77.1 \pm 4.2$ | $59.8 \pm 2.7$ | $46.6 \pm 4.1$ |
| Fraction as $CO_2$ (%) | 34 | 59 | 48 | 33 |
| Fraction as $CH_4$ (%) | 36 | 25 | 32 | 41 |
| Fraction as $N_2O$ (%) | 29 | 16 | 21 | 26 |
| Difference from conventional baseline |  | $p < 0.05$ | NS | NS |

*Data are means ± 1 std. dev

dairy, lowland grazing livestock, Less Favoured Area (LFA) grazing livestock, pigs and poultry, and other. These cover the entire agricultural land-base in England and Wales. The Objective Function of the model, which is maximised subject to constraints on resource availabilities, is the sum of total crop and livestock production, expressed as ME. Although human diets also need proteins, fats and nutrients, energy requirements are deemed to be a primary driver of consumption and an inadequate food-energy intake is almost always accompanied by insufficient intake of nutrients[37].

The basic formulation of the OLUM is

$$Z = \sum_{ij=0}^{n} C_{ij} \cdot x_{ij} \text{ subject to } Rx_{ij} \leq \mathbf{b}, x_{ij} \geq 0, \qquad (1)$$

where $Z$ is the objective function to be maximised, $C_{ij}$ is the ME output (fresh weight per unit crop area or livestock number $yr^{-1}$) of agricultural product $i$ on soil × rain class $j$, $x_{ij}$ is a scalar for the agricultural activity (crop area or livestock number), $Rx_{ij}$ is a factor for the input and resource requirement associated with the agricultural activity, and $\mathbf{b}$ is a vector for resource endowment and input availability (e.g., land by soil and rainfall class, and available soil N). Human dietary change is not considered.

In each farm type, the set of crop and livestock production activities available are fixed, as evidence suggests that the dominant agricultural activity (e.g., dairy farming) will usually stay in place post conversion to organic management, due to existing farm infrastructure, farming knowledge and local conditions[43]. However, these activities can be individually expanded and contracted endogenously. The land areas under each farm type are fixed, reflecting the areal coverage of their conventional equivalents recorded in the June Survey of Agriculture in 2010[44]. A number of logical constraints are applied in the model to reflect: the availability of land in the various soil/rainfall classes (next paragraph); maximum permissible area of crop groups (e.g., cereals, root crops) reflecting rotational constraints; and upper limits on the total output of each crop, set at 150% of the current supply, following an assumption that further increases could not be absorbed by the market. Rotational N availability limits are also imposed, as determined by crop and livestock-product offtake (from the land), N supply from various sources, such as biological fixation, imported feed and atmospheric deposition, as well as manure-N availability in each region. We assume balances of P and K are maintained by applying P and K minerals commonly used in organic systems. Livestock numbers and associated product output volumes are constrained by feed availability, as well as maximum and minimum stocking density constraints.

Heavy, medium, light and humose soil classes are defined with specified organic matter contents and pH values, and their spatial distribution across England and Wales in 5 km × 5 km grid squares were obtained from the National Soil Inventory (www.LandIS.org.uk). Four rainfall classes are defined based on 30-year Meteorological Office annual rainfall data: dry 539–635 mm, medium 636–723 mm, wet 724–823 mm and very wet 824–2500 mm. The total areas of each soil × rainfall combination were determined by identifying the dominant combination in each 5 km × 5 km grid square and allocating to that combination the sum of the areas of each square, less any non-agricultural area.

The OLUM produces a best estimate of production under fully organic agriculture in England and Wales, assuming that food production would be maximised. To ensure that the results are reasonable, outputs are compared to the real-world distribution of conventional production in 2010 derived from a range of industry sources (Supplementary Table 4), and to results from a previous study on the production impacts of a switch to organic farming in England and Wales[15].

**The Agri-LCA models**. We assessed the environmental impacts of conversion to organic farming using the Cranfield Agri-LCA models for England and Wales[3]. Fossil energy use and emissions of $CO_2$, $CH_4$ and $N_2O$ per tonne of each food commodity produced under given soil and management conditions are combined with official data on levels of production, to provide estimates of the total GHG impact of agriculture. Results from earlier emissions analyses generated by these models for the current mix of agricultural systems in England and Wales are used as a comparator against which to assess the organic conversion scenario. We adjust the following components of the Agri-LCA models to better reflect organic agriculture using data sources listed in Supplementary Table 4: first, crop and grassland yields; second, crop cultivation practices and manure/compost application rates; third, crop and grassland areas by soil and rainfall type; fourth, livestock productivity and mortality rates; and fifth, livestock diet compositions.

Crop yield, cultivation and manure application data are adjusted for 12 main crops: wheat, barley, rye, oats, potatoes, oilseed rape, sugar beet, beans and peas, cabbage, carrots, onions and forage maize. These cover 98% of the cultivated land in England and Wales[44]. All data sources used in this exercise are provided in Supplementary Table 4. Crop and grassland areas under each of 16 soil and rainfall classes are derived from the OLUM results. The crop areas, by each soil and rainfall class, are used in the Agri-LCA models to adjust $N_2O$ and $CO_2$ impacts to reflect organic management. The functional units used in the LCA are tonnes of marketed crop-product.

Organic animal production data for the Agri-LCA are drawn from a range of industry sources to define, by livestock type: daily live-weight gain, annual fat-corrected milk yield, and feed conversion ratios. Data are also input to the Agri-LCA on the composition of livestock diets, stocking rates per hectare and the

proportion of livestock on upland and lowland. These values ensure that feed intake meets the ME demand of livestock. Nitrogen excretion from livestock is derived from mass balances. Compound feed composition data are also applied to determine embedded impacts of feed production overseas. Direct $CH_4$ emissions from livestock are calculated as a function of dry matter intake (scaled in proportion to the forage dry matter intake), live-weight and milk yields. The Agri-LCA livestock emissions estimates are based on six commodities: eggs, milk, sheep, beef, pig and poultry meat. Meat outputs are defined in terms of total dressed carcass weight (tonnes), eggs by weight (tonnes) and milk output as fat-corrected litres[3].

**System boundaries and allocation of environmental burdens**. The downstream system boundary applied in the Agri-LCA modelling is the farm gate, i.e., only resources consumed during the production of inputs and on-farm-based processes are considered (i.e., 'from cradle to farm gate'[2]). The GHG emissions associated with downstream activities—such as distribution, consumption and disposal of products produced on the farm—are not included. Some on-farm processing, such as grain drying, milk cooling and potato storage, are included in the total impact assessment, as these operations are considered to be part of the on-farm production process[3]. Where multiple products are derived from the same agricultural activity, such as grain and straw from cereals production, the GHG emissions from fossil energy use associated with the different components are allocated on the basis of relative economic value and by system expansion with regard to manure (i.e., the manufactured N fertiliser avoided is discounted from the environmental burdens associated with non-organic crops). Where economic allocation is used in the Agri-LCA, an organic price differential is applied. Emission factors are derived from IPCC 2006 estimates and total emissions of $CH_4$ and $N_2O$ converted to $CO_2$ equivalents using their 100-year Global Warming Potentials (GWPs). The time-dependency of the GWP values introduces some uncertainty, particularly for $CH_4$ which has a 20-year GWP more than twice its 100-year value. However, allowing for this would introduce undue complexity. The emissions associated with animal feed production are allocated to the livestock emission estimates, not those for crop production.

**Imports and exports**. The GHG emissions associated with producing imported food are allowed for in the Agri-LCA models. We assume that any shortfall in supply from organic agriculture is made up by increased imports of organically produced commodities from overseas. We use data from industry sources (Supplementary Table 5) to allocate imported product to the historic regions of origin of imports[45]. The GHG emissions associated with the transport of imports to England and Wales is determined by multiplying the total volume of imported products by GHG coefficients derived from Hess et al.[45]. Transport burdens for imported sugar and sheep meat are derived from Plassman et al.[46] and Webb et al.[23], respectively.

Where the OLUM generates crop and livestock production in excess of domestic demand, the surpluses are assumed to be exported and the GHG and fossil energy burdens associated with production of the exported commodities are subtracted from the total environmental burdens of organic agriculture. The same adjustment is made to the GHG estimates of exports for conventional agriculture (see data sources for export volumes in Supplementary Table 5). Where the OLUM reduces production below the level of domestic demand it is assumed that no exports occur, i.e., domestic consumption would take priority.

Fossil energy use and GHG emissions associated with the production of oilseed rape, sugar beet, wheat and lamb from non-European countries are derived from Pelletier et al.[47], Tzilivakis et al.[48] and Webb et al.[23]. The environmental burdens associated with crop and livestock products sourced from Scotland, Northern Ireland and the rest of Europe are derived from the Agri-LCA, under the assumption that similar emissions and fossil energy use would occur in these systems[3].

**Soil carbon sequestration**. We obtain an upper estimate of potential sequestration rates in organic systems based on rates of change of soil C measured in the National Soil Inventory (NSI) of England and Wales[49], as follows. Kirk and Bellamy[28] summarised the NSI results by fitting to the data the simple single-pool model

$$dC/dt = I - kC, \qquad (2)$$

where $C$ is the C content per unit land surface area, $I$ is the rate of input from vegetation and other sources and $k$ is a rate constant for decomposition. They fitted Eq. (2) to the data for each NSI land use class separately, omitting organic soils (which accounted for <5% of all the soils in the NSI) because their rates of change were less certain. The soil C content at steady state, when $dC/dt = 0$, is equal to $I/k$. Soils with C content greater than the steady-state value lose C; those with C contents less than it sequester C.

We take the NSI class 'rotational grass' (i.e., grass that is sown and then tilled every few years as part of an arable rotation) to represent potential C contents under ideal organic management, and the class 'arable' to represent C contents under conventional arable management. The mean soil C contents were 43.2 ($n = 552$ sites) and 58.7 ($n = 301$ sites) Mg C $ha^{-1}$ under arable and rotational

grass, respectively, and the calculated steady-state C contents were 37.6 and 55.0 Mg C ha$^{-1}$, respectively, indicating the rotational grass soils were on average close to steady state and their C contents therefore represent maximum potential sequestration levels. The values of $I$ and $k$ for rotational grass were 2.54 Mg C ha$^{-1}$ yr$^{-1}$ and 0.046 yr$^{-1}$, respectively (equivalent to negative emissions of $-9.3$ and $-0.17$ Mg CO$_2$ ha$^{-1}$ yr$^{-1}$). Substituting these values and the mean arable C content in Eq. (2) gives for the mean rate of sequestration on conversion from arable to rotational grass $((2.54 - 0.046 \times 43.2) + 0)/2 = 0.28$ Mg C ha$^{-1}$ yr$^{-1}$ (or $-1.03$ Mg CO$_2$ ha$^{-1}$ yr$^{-1}$). After adjusting for the proportion of arable to arable plus rotational grass, the rate is $0.28 \times 552/(552 + 301) = 0.18$ Mg C ha$^{-1}$ yr$^{-1}$ (or 0.66 Mg CO$_2$ ha$^{-1}$ yr$^{-1}$). We use this as the high C sequestration rate in Fig. 3. We assume sequestration rates in established swards of permanent pasture or rough grazing to be zero given that these sites will have already reached steady state.

For comparison, in a literature survey of experiments comparing conventional and organic farming, Gattinger et al.[24] found sequestration rates between 0.07 and 0.45 Mg C ha$^{-1}$. However, most of these comparisons involved very high rates of external organic matter inputs to the organic systems. The average inputs were four times those under conventional farming for the full dataset and two times for systems with inputs equivalent to those from one European Livestock Unit (ELU) ha$^{-1}$ [26]. We calculate that quantities of farmyard manure would be only approximately 12% greater under organic farming, as a result of greater numbers of grazing livestock (Supplementary Table 3). We therefore consider Gattinger et al.'s upper and middle sequestration estimates to be unrepresentative and take as the moderate sequestration rate in Fig. 3 their lower value of 0.07 Mg C ha$^{-1}$ yr$^{-1}$.

Gains through C sequestration will be time-limited, because any given soil has a finite capacity to accumulate C and a new steady-state C content will be reached after a few years, when increased C inputs are matched by increased losses at the greater soil C content. Our estimated sequestration rates therefore only apply in the early-years following conversion to organic methods. Based on the NSI data, a new steady-state C content on conversion from arable to rotational grass would only be attained after $(55.02 - 43.15)/0.28 = 42$ years.

**Additional emissions from overseas LUC and C sequestration.** We estimate the additional overseas land area required for each of the food products listed in Fig. 1, produced organically, as follows. For crops, we use first, regional yield data from Eurostat, second, organic crop yields from the recent meta-analysis by de Ponti et al.[32] and third, results of an LCA for milling wheat grown in Canada[47]. For livestock, we use first, regional yield data from Eurostat, second, results from the Agri-LCA[3] and third, recent studies on the environmental burdens of imported lamb from New Zealand[23,50]. The additional land area is calculated from the total overseas area required less the amount required for imports in the conventional baseline (based on the values in Supplementary Table 6). The corresponding emissions are calculated as follows.

We assume that woodland would not be converted for food production as this would conflict with the principles of the International Federation of Organic Agriculture Movements (IFOAM)[51]. We calculate emissions from the conversion of grassland to crops from the area converted multiplied by LUC emission estimates specified by the British Standards Institute for a range of countries[52]. Considering that not all the LUC is from grassland, we compare three ways of assessing the net emissions from overseas LUC and associated soil C sequestration, plus that of home production, as follows. First, High: all the additional land required is converted from grassland, with no net soil C sequestration at home or overseas. Second, Medium: 50% of the additional arable land is converted from grassland, with a moderate rate of C sequestration (0.07 Mg C ha$^{-1}$ yr$^{-1}$) at home and overseas. Third, Low: 25% of the additional arable land is converted from grassland, with a high rate of C sequestration (0.18 Mg C ha$^{-1}$ yr$^{-1}$) at home and overseas.

Following Searchinger et al.[35], we also calculate the additional 'carbon opportunity cost' (COC) of using the land for agriculture as the quantity of C that could be sequestered annually if the average productive capacity of land used to produce 1 kg of each food product globally were instead devoted to regenerating forest. We calculate the total COC from Searchinger et al.'s[35] COC factors per unit fresh weight of each food product (separating crops for human consumption from those used as animal feeds) multiplied by the additional fresh weight imports of each product required to offset home production shortfalls. This is in addition to the emissions calculated under the LUC and C sequestration scenarios (1)–(3) above. This 'C gain' method—as opposed to a 'C loss' method based on plant and soil C lost to date per unit food production—applies if it is only possible to increase C by re-establishing forests.

**Uncertainty analysis.** Estimates of uncertainty for each main commodity analysed were produced following the method of Wiltshire et al.[53]. Uncertainties were derived using Monte Carlo simulations with each domestically produced crop commodity given an uncertainty estimate of 10% (i.e., in a triangular distribution with upper and lower bounds at 10% of the mean) and each domestically produced livestock commodity at 15%. The emissions for crops and livestock were summed in separate Monte Carlo simulations to produce overall uncertainty estimates for each sector (as the standard deviation). These were increased by 15% for all imported commodities en bloc. Emissions from import transportation were

assumed to have a standard deviation of 10% of the mean[53], i.e., the coefficient of variation (CV) is 10%. The areas of land derived by the LP were assumed to have an error of 15%, which was applied to the whole solution, not per crop, given that all areas were derived from any individual solution. Error bars on production area per crop (or livestock commodity) are thus not shown.

The final emissions and uncertainty estimates for each production system were derived from the sum of emissions from domestically produced crops and livestock together with emissions from imported crop and livestock production, together with their transport emissions, based on supply chain data from Webb et al. (2013)[23] and Williams et al. (2017)[54]. Estimates of the uncertainty from LUC were derived from Houghton[55] and those from C sequestration from Kirk and Bellamy[28] for the upper rate and from Gattinger et al.[24] for the medium and lowest rates. These were implemented as the uncertainty being a proportion of the means that were applied to the LUC and C sequestration scenarios. These were established as having a CV of 17% for LUC[55], which was increased for the carbon opportunity cost of Searchinger et al.[35] by a factor of 1.5 to allow for the extra uncertainty of the method (i.e., CV of 26%). The uncertainty of the high level of C sequestration was 86 and 24% for lower levels.

The uncertainty estimates for the sum of crop and livestock commodities, transport and land use change emissions and sequestration are summarised in Supplementary Table 7. These were used as input values of uncertainties in the last stage to derive the overall uncertainties of each scenario. We tested the significance of differences in mean values, $z$, using Eq. (3)[53]

$$z = \frac{|m_A - m_B|}{\sqrt{CV_A^2 \times m_A^2 + CV_B^2 \times m_B^2}} \times 100, \qquad (3)$$

where $m_A$ and $m_B$ are the means of systems A and B, respectively, and CV is the CV of each mean (expressed as whole numbers). The threshold for a significant difference at the 5% level was $z \geq 1.96$.

**Reporting summary.** Further information on research design is available in the Nature Research Reporting Summary linked to this article.

## Data availability

The data underlying these calculations can be accessed at: https://doi.org/10.6084/m9.figshare.6080333.v2. OLUM model code and data can be accessed at: https://tinyurl.com/yxlszsrv. The Agri-LCA models and data can be accessed at: https://tinyurl.com/yy5jol7c

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

## Acknowledgements

We thank the Soil Association, Vitrition Organic Feeds and AHDB for providing data for the land-use model. L.S. was supported by the Organic Research Centre, a PhD studentship from the Engineering and Physical Sciences Research Council (EPSRC grant ref. WG17023N) and an education grant awarded by the Ratcliff Foundation. We thank Dr. Bruce Pearce at the Organic Research Centre for his comments on an early draft of the paper.

## Author contributions

L.S., P.J., A.W. and G.K. designed and developed the OLUM and LS carried the model simulations. A.W. developed the Agri-LCA models, provided an overview of their function for use in this study and access to relevant data for the calculation of the environmental impacts. G.K. provided guidance on the calculation of the greenhouse gas emission offset from carbon sequestration in organic systems. L.S. and G.K. wrote and revised the paper with help from all co-authors.

## Competing interests

The authors declare no competing interests.
