## [Peer Review File · Nature Communications]

Reviewers' comments:

Reviewer #1 (Remarks to the Author):

The authors use the result of a previous study in press to compare to national GHG budgets of England and Wales, which is specified as the main novelty. They thus provide the most up-to-date study in the UK. The main finding, that a change to organic production is found in earlier studies as the y cite (e.g. 17-19), but not on the system level for the UK.

General concerns:

The results are presented without the proper uncertainty estimates. agricultural emissions and practices are highly uncertain and thus all the numbers presented must be considered with great care regarding robustness, which is missing so far (i.e. ruminants having lower GHG emissions probably depends on many assumptions made - as they are also for pigs).

Optimizing for edible energy is a questionable goal, since diets also need proteins, fats and other nutrients. Why should the UK optimize for energy (low quality nutrients) and not for proteins or a mixed diet, if the rest needs to be imported anyway (i.e. no diet change is considered?).

A critical discussion of the results and analysis is limited and the conclusions are basically referring to other publications.

The reproducibility of the research is very limited, since the information provided (incl. SI) is not really strong.

Detailed comments:

System boundaries exclude food related emissions after the farm. Is this consistent with the calculations of imported food (i.e. is the additional transport considered?) Also, is the increased price for organic products included in the allocation of co-products (compared to the cited study)?

I think the literature cited is not most balanced, since many references are self-citation, even when not necessary (e.g. the reference used for LCA is definitely not the most appropriate).

Carbon sequestration: indeed this is very uncertainty and using scenarios is fine. the authors mention "We highlight that the above carbon sequestration gains will be time-limited, as any given soil will have a finite capacity to accumulate C and a new steady state will be reached after a period of time". I don't see this highlighted in the results/discussion and how is it addressed (i.e. how is the time limitation taken into account)? Are there 2 scenarios?

In the conclusion part the authors say "Our analysis suggests enhanced soil C sequestration in organic systems could potentially offset 4-38% of GHG emissions from domestic and overseas food production, through use of leys and organic manures and composts. " However, based on the methods it should be 0-38% and it should be mentioned that 4-38% can be achieved in the first 20 years and 0 after (based on the methods).

external land use. (figure 4). it seems quite surprising that currently only a small share of food is imported from outside the. Does this include sugar cane, rice, cotton, tea and coffee?

Land use change

The GHG for land use change seems to be really rough and there is more detailed assessment for different regions (which could be applied to the respective production regions in organic scenarios). How sensitive are these?

In the sensitivity, the authors assume only 25 or 50% being transformed for pasture to organic cultivation. does this involve that people would need to hunger or change diets? Why is there no scenario for assuming it is converted from forests as a sensitivity assessment (like 50%). Palm oil or soy bean are usually not grown on grassland.

-> as analyzed, the effect is mainly important for oilseed assessment, but then it should be included in the optimization (i.e. give fats and proteins higher weight than carbohydrates).

The figures could be improved to enhance information density.

Supp table 1-2: Is there really only one reference used per item?

Supp table 3: how comes the estimate by Williams are so low for overseas beef and sheep production compared to other sources? What are these other sources?

Reviewer #2 (Remarks to the Author):

General comments

The study addresses the highly contemporary question of consequences of a conversion of food production from conventional to organic farming for GHG emissions. It does so in a comprehensive, intelligent and appealing way by combining a land use model with a life cycle assessment. Both types of models have previously been published, but are extended with considerations of land use change and soil carbon sequestration, which are essential for answering the question. In these respects, this study clearly goes beyond other published studies.

There is throughout the manuscript a confusion between food production and food consumption. As I understand the land use model maximizes food production for the territory of England and Wales. I therefore assume that the values in Figure 1 reflect the food (crop and livestock) production in this region. However, it is more unclear what Figure 3 actually shows. The figure caption and the manuscript text mentions that this refers to food production; however, METHODS mentions (lines 368-369) that GHGs were calculated to meet domestic food demand. This is also indicated on lines 54-55. Therefore, Figure 3 rather reflects the GHG emissions from food consumption in E&W. The manuscript does not describe how food consumption is quantified and how the production and import are adjusted to meet this food consumption. This confusion should clearly be corrected.

The manuscript makes the conclusion that reductions in GHGs would only be achievable by major changes in UK national diet. This is jumping to conclusions. The study does not investigate this issue or even other alternatives, such as emissions reduction technologies in both conventional and organic systems.

The study integrates GHG emissions of primarily methane and nitrous oxide from agricultural systems with CO₂ emissions from carbon stock changes (soil and vegetation). This is highly problematic (although too often done) due to different time perspectives, as also to a limited extent discussed in the manuscript. Carbon stock changes from land use change and soil management have a short time perspective (typically 20 years). Here they are compared with other GHG emissions with a 100-year time perspective (100 year GWP). These values are in reality not comparable. There is no easy solution. However, I suggest to raise this perspective in the paper. It could be done by including two more bars in Figure 3, where bars in graphs a and b could be supplemented with GHG emissions from methane, nitrous oxide and fuels using a 20-year GWP.

I find this study very interesting, of high quality and I favor publication, considering my comments .

Specific comments

Lines 92-93

Why are these land-use changes so much larger than in the global study?

Line 288

Where are these outputs presented?

Response to Reviewers' comments

We thank the reviewers for their very helpful comments. We have addressed all the points raised and provide a point-by-point response below.

Reviewer 1

General

1. *The results are presented without the proper uncertainty estimates. agricultural emissions and practices are highly uncertain and thus all the numbers presented must be considered with great care regarding robustness, which is missing so far (i.e. ruminants having lower GHG emissions probably depends on many assumptions made - as they are also for pigs)*

We accept this was a weakness of the original MS. We have made uncertainty estimates as described in Methods (lines 460–472) and added these to the results for GHG emissions (Figs 2–3 and new Table 1). The method we used involved assessing the ranges of values for the different commodity, management and land use variables in our calculations, and running Monte Carlo simulations to obtain means and standard deviations of the results. This is the most realistic approach given the wide range and large number of variables inherent in our calculations.

2. *Optimizing for edible energy is a questionable goal, since diets also need proteins, fats and other nutrients. Why should the UK optimize for energy (low quality nutrients) and not for proteins or a mixed diet, if the rest needs to be imported anyway (i.e. no diet change is considered?)*

Although human diets also need proteins, fats and nutrients, energy requirements are a primary driver of consumption and an inadequate food-energy intake is almost always accompanied by insufficient intake of nutrients. Metabolisable energy is therefore the single most informative measure of food production. We have not considered dietary change as the focus of the OLUM, and the Agri-LCA is based on production rather than consumption. We have added a clarifications of these point in Methods (lines 280–290 and lines 326-344).

3. *A critical discussion of the results and analysis is limited and the conclusions are basically referring to other publications.*

We have considerably expanded the discussion of the results and referred to a broader range of publications. We have more than doubled the length of the main text (2,914 vs 1,353 words).

4. *The reproducibility of the research is very limited, since the information provided (incl. SI) is not really strong.*

We have expanded the list of references in Methods and SI. All the underlying data and models are freely available to readers as indicated in Data availability.

Specific

5. *System boundaries exclude food related emissions after the farm. Is this consistent with the calculations of imported food (i.e. is the additional transport considered?)*

The system boundaries are defined as ‘cradle-to-farm-gate’ consistent with previous LCA studies (see lines 345–357).

6. *Also, is the increased price for organic products included in the allocation of co-products (compared to the cited study)?*

Where economic allocation was used in the Agri-LCA, we applied an organic price differential. We have added this clarification in Methods (lines 356–357).

7. *I think the literature cited is not most balanced, since many references are self-citation, even when not necessary (e.g. the reference used for LCA is definitely not the most appropriate).*

We have expanded the list of references, particularly for LCA with Poore & Nemecek (Science 360, 987–992, 2018), Nemecek et al. (Agric. Syst. 104, 217–232, 2011) and Van der Werf et al. (Agric. Ecosyst. Environ. 118, 327–338, 2007).

8. *Carbon sequestration: indeed this is very uncertainty and using scenarios is fine. the authors mention "We highlight that the above carbon sequestration gains will be time-limited, as any given soil will have a finite capacity to accumulate C and a new steady state will be reached after a period of time". I don't see this highlighted in the results/discussion and how is it addressed (i.e. how is the time limitation taken into account)? Are there 2 scenarios?*

We have added a new section ('Extent of soil carbon sequestration', lines 129–148) in which we make clear our carbon sequestration estimates only apply to the first 10–20 years after conversion to organic methods. We also explain in the new section 'Extent of overseas land conversion' that soil carbon losses arising from conversion of pasture to crops overseas (to offset the shortfall in home production) would occur over a similar 10–20 year time-frame, and so could potentially be offset by soil carbon conserving or sequestering measures under organic production (lines 149–159). However the net change in emissions will be sensitive to the nature of the land use change and the availability of suitable land, as we explain in this section.

9. *In the conclusion part the authors say "Our analysis suggests enhanced soil C sequestration in organic systems could potentially offset 4-38% of GHG emissions from domestic and overseas food production, through use of leys and organic manures and composts. " However, based on the methods it should be 0-38% and it should be mentioned that 4-38% can be achieved in the first 20 years and 0 after (based on the methods).*

See response to Point 8. We have revised this statement to make this clear (lines 136–143).

10. *External land use. (figure 4). it seems quite surprising that currently only a small share of food is imported from outside the. Does this include sugar cane, rice, cotton, tea and coffee?*

The external land use change figures only include land that is required to produce the food products listed in Figure 1. Food products that aren't produced in the UK on a large scale, such as maize, rice, tea, coffee, sugar cane etc., are excluded. We have amended the Figure 4 caption and Methods (414–415) to make this clearer.

11. *Land use change. The GHG for land use change seems to be really rough and there is more detailed assessment for different regions (which could be applied to the respective production regions in organic scenarios). How sensitive are these?*

We have applied emission estimates for regional land use changes based on the PAS2050 values of the British Standards Institute with maximum and minimum values for each region (lines 425–436). We have applied the maximum and minimum values to Figure 4 to illustrate the range of results. We have also added the total land-use overseas to Figure 4 and have included error bars on the GHG emissions from land use change estimates following the methods outlined under Point 1.

12. *In the sensitivity, the authors assume only 25 or 50% being transformed for pasture to organic cultivation. does this involve that people would need to hunger or change diets? Why is there no*

scenario for assuming it is converted from forests as a sensitivity assessment (like 50%). Palm oil or soy bean are usually not grown on grassland.

We assume that woodland would not be converted to arable land for organic production as this would release considerably more CO₂e than tillage of pasture and would represent a direct conflict with the International Federation of Organic Agriculture Movements (IFOAM) organic principles (lines 428–429).

13. As analyzed, the effect is mainly important for oilseed assessment, but then it should be included in the optimization (i.e. give fats and proteins higher weight than carbohydrates).

We have maintained a production and dietary balance based on current consumption patterns using multiple constraints in the OLUM (e.g. to avoid over-production of crops with a high ME content and to avoid the implementation of farming systems that are not already represented in real-world organic approaches to production).

14. The figures could be improved to enhance information density.

We have added error bars to the figures to improve information density and uncertainty evaluation.

15. Supp table 1-2: Is there really only one reference used per item?

We have added references to additional datasets in the supplementary material

16. Supp table 3: how comes the estimate by Williams are so low for oversea beef and sheep production come pare to other sources? What are these other sources?

The land area requirements from ‘other literature’ in Supplementary Table 3 used yields from DePonti et al. (Ref 23) which were lower than those used by Williams et al. (Ref 2). We have clarified this in the table caption and added to the list of data sources in the SI.

Reviewer 2

General

1. There is throughout the manuscript a confusion between food production and food consumption. As I understand the land use model maximizes food production for the territory of England and Wales. I therefore assume that the values in Figure 1 reflect the food (crop and livestock) production in this region. However, it is more unclear what Figure 3 actually shows. The figure caption and the manuscript text mentions that this refers to food production; however, METHODS mentions (lines 368-369) that GHGs were calculated to meet domestic food demand. This is also indicated on lines 54-55. Therefore, Figure 3 rather reflects the GHG emissions from food consumption in E&W. The manuscript does not describe how food consumption is quantified and how the production and import are adjusted to meet this food consumption. This confusion should clearly be corrected.

The OLUM maximises food energy production in England and Wales and the GHG estimates were calculated in line with the modelled levels of domestic production in addition the overseas production required to meet current levels of national consumption in England and Wales for the crops and livestock products listed in Figures 1 and 2, i.e. we did not consider overseas land use for imports of food not produced at home. We have added clarification on these points (see response to Reviewer 1, Point 10).

2. The manuscript makes the conclusion that reductions in GHGs would only be achievable by major changes in UK national diet. This is jumping to conclusions. The study does not investigate this

issue or even other alternatives, such as emissions reduction technologies in both conventional and organic systems.

We accept this criticism and have re-written and broadened the Discussion to include other possible ways to reduce emissions.

- 3. The study integrates GHG emissions of primarily methane and nitrous oxide from agricultural systems with CO₂ emissions from carbon stock changes (soil and vegetation). This is highly problematic (although too often done) due to different time perspectives, as also to a limited extent discussed in the manuscript. Carbon stock changes from land use change and soil management have a short time perspective (typically 20 years). Here they are compared with other GHG emissions with a 100-year time perspective (100 year GWP). These values are in reality not comparable. There is no easy solution. However, I suggest to raise this perspective in the paper. It could be done by including two more bars in Figure 3, where bars in graphs a and b could be supplemented with GHG emissions from methane, nitrous oxide and fuels using a 20-year GWP.*

We calculate the CO₂ equivalent-impacts of CH₄ and N₂O emissions allowing for their different Global Warming Potentials (GWP) integrated over 100-years, as is conventional. We agree this introduces some uncertainty, particularly for CH₄ which has a 20-year GWP more than twice its 100-year value. However there is no easy solution to this and we adopt the 100-year convention following the IPCC. We added a comment on this (lines 358–361). No such correction applies to annual CO₂ emissions because CO₂ has unit GWP by definition, regardless of the integration time. The question of how to deal with the time-dependency of enhanced soil carbon sequestration under organic methods is separate. We have changed the text to make clear that our estimates of sequestration only apply to the first 10–20 years following conversion, after which they fall to zero (lines 379–387).

Specific

- 4. Lines 92-93 Why are these land-use changes so much larger than in the global study?*

More intensive production systems are present in the UK which leads to a higher land-use penalty following a switch to organic systems (lines 154–157).

- 5. Line 288 Where are these outputs presented?*

We have added a cross reference to the correct Figure (Fig. 1) in Methods (line 308).

Reviewers' comments:

Reviewer #1 (Remarks to the Author):

General comments:

As it is known by now that organic is not better for the climate, maybe mention that other benefits might occur (i.e. if it is climate neutral it might be the better choice). Also, what are the implications of higher potato production on other environmental impacts. Isn't organic potato involving lots of copper application that damages soil fertility? Maybe discuss some effects beyond GHG.

The paper contains several typos (e.g. in line 31 and 382)

While I understand optimizing over other aspects than edible energy, it would still be good to show the summary of nutritional content for the organic vs conventional scenarios to give a bit more content to the paper

Detailed comments:

L33-36. I read that the study of 2009 used LCA data from the study of 2018. This seems to be a mistake.

Results:

L 68: What are exactly the limits and how have they been selected?

L70: Does it really make sense to grow sugar beet instead of cereals from an environmental and food perspective?

L86: I understand this might lead to higher N₂O emission, but these are also highly uncertain. This should be considered in the uncertainty assessment.

L170ff: I think this is already discussion

Discussion:

L 178-179: This sentence seems a bit weird ("improvement or otherwise in...")

L190ff: this is only limited for GHG, for other aspects (ecosystem services it is not limited to 20 years)!

L210: Why less meat is generally good. It needs to be put in context to the results, where you mention that pasture animals might be an option (I think also the N and P cycling might benefit from some livestock).

It is not really clear how N and P limitations have been modeled. There is some legumes, but what about P?

Figures:

Fig. 1 It should be noted what OSR is. The multiplier for the number scale is a bit odd. Especially it is not clear if milk numbers are related to production or livestock number scale.

Fig 2. N₂O seems to dominate the impacts, while usually it is around 10% of total impacts on a global level - what are the reasons?

L237 "not allowed for". Does it mean it is not shown or it is excluded from the analysis? In next

figures you show it.

Fig 3. If you have extensive figure captions, also highlight these numbers only refer to the first 20 years after conversion

Table 1: Split the results also per CH₄, CO₂ and N₂O, to show the total changes

Methods:

L281. at least calculate the total macro nutrients in the two cases.

L331: Might there be an issue with N₂O modeling in the use model?

L398: How exactly are the overseas GHG calculated?

L431: What about the ripple effect. This means someone else has to cut down the forest and then the organic farmer can transform it from pasture to cropland (and new pastures will be created?) I think this is a big assumption and similar to the assumption that I am not responsible for the GHG emission of the plane, since it is scheduled to fly anyway. At least it should be discussed that indirectly organic agriculture can be responsible for deforestation).

Whole uncertainty section.

1- It is not fully clear to me and I expect to have a table with the details in the SI.

2- It seems to be really low uncertainty (+/- 15% in agriculture is really low). Also +/-1% for transport is probably even not covering the same company and transport mode (due to wind etc.)

Reviewer #2 (Remarks to the Author):

General comments

The authors have adequately dealt with my previous comments. However, with the revised manuscript two issues become clearly prominent (the change in SOC and the effects of LUC). For both issues, I have some additional concerns that need to be addressed before I can recommend publication.

The calculation of the effects on SOC relies solely on the estimates of Gattinger et al. (2012). This is highly problematic, not least due to the reasons described by Leifeld et al. (2013; PNAS 110:E984), which criticized the Gattinger paper. Essentially the meta-analyses of Gattinger do not compare equivalent farming systems, as seen from a systems perspective. The organic farms in the meta-analysis had in many cases a higher external carbon import (e.g. of manure and compost) than compared conventional systems. In fact, they mostly compared mixed organic systems with mineral fertilizer based conventional systems. Whereas such comparisons may be valuable from a scientific perspective, they cannot be applied to represent the broader agriculture from a systems perspective. I think it is worth quoting Leifeld et al. (2013) on other problems with the meta-analysis that is also relevant to the use of these data in the current context: "In most situations, manure would be applied somewhere anyhow, unless used for other purposes. Hence, an increase in soil C in one field (whether in an OF or CF system) is not a net transfer of C from atmosphere to land but a movement of C from one site to another". The current study shows that overall livestock production would decline in the organic farming scenario compared to the conventional one. Therefore, there cannot be an increased amount of manure or compost to be applied on the organic farmed land. This aspect of manure as a mitigation measure was also noted by Powlson (2011; European Journal of Soil Science 62:42–55). The authors need to revise their calculations taking the issue of the available resources of manure and compost into consideration.

Regarding LUC there is a newly published study by Tim Searchinger (2018, Nature 565:249-253) that in a systematic way addresses the GHG impact of LUC by introducing the concept of opportunity cost. It would be highly pertinent within the current manuscript to make reference to this newly published study and to put the applied method into perspective of the methodology proposed by Searchinger et al. (2018).

Specific comments

Line 128

Change "decreased" to "reduced".

Lines 144-148

These text should be deleted. It does not present results of the analysis. The comments are outside the scope of the study, and in fact the statements can be challenged.

Line 192

I would really challenge the concept that application of manure and composts can increase SOC in organic farming. This requires that these materials can be sourced. Figure 1 shows that overall livestock production is lower in the organic farmed scenario than for the conventional scenario. How can there then be more manure applied? The same concern applies for compost?

Line 312

Delete "an".

Lines 379-381

I suggest deleting this sentence. I would consider this self-evident.

Authors' response to Reviewers' comments

We thank the reviewers for their very helpful comments. We have re-written large parts of the MS in the light of the comments and made additional calculations. We have addressed all the points raised, as follows.

REVIEWER 1

1. *As it is known by now that organic is not better for the climate, maybe mention that other benefits might occur (i.e. if it is climate neutral it might be the better choice). Also, what are the implications of higher potato production on other environmental impacts. Isn't organic potato involving lots of copper application that damages soil fertility? Maybe discuss some effects beyond GHG.*

We have added discussion of other potential benefits from organic farming, and their significance at a global scale (lines 201-208).

2. *The paper contains several typos (e.g. in line 31 and 382)*

Apologies. We have carefully proof read the revised MS.

3. *While I understand optimizing over other aspects than edible energy, it would still be good to show the summary of nutritional content for the organic vs conventional scenarios to give a bit more content to the paper*

We have commented on the levels of protein production, referring to our earlier study (Smith et al. 2018, Ref 16) (lines 65-66 and 72-73) and added data from Eatwell Group (Supplementary Table 2), compared to the conventional level of production.

4. *L33-36. I read that the study of 2009 used LCA data from the study of 2018. this seems to be a mistake.*

This was an error. We have deleted the erroneous reference Poore & Nemecek (2018).

5. *L 68: What are exactly the limits and how have they been selected?*

Upper limits were set at 150% of the current supply, on the basis that further increases could not be absorbed by the market and evidence that most consumers are unwilling to make major changes to diet. We have made this clearer (lines 79-80).

6. *L70: Does it really make sense to grow sugar beet instead of cereals from an environmental and food perspective?*

Total sugar beet production decreased, but, due to its high ME yield, it reached its upper local limit in parts of eastern England. These limits were imposed to restrict expansion of production into areas too away from major processing, i.e. where transport costs would be prohibitive. We have made this clearer (lines 80-82).

7. *L86: I understand this might lead to higher N₂O emission, but these are also highly uncertain. This should be considered in the uncertainty assessment.*

We have clarified in the Methods that N₂O (and CO₂ and CH₄) emission estimates are tailored to local soil and management conditions (lines 263-270 and 282-286). The main uncertainty in emission estimates is therefore the uncertainty in the drivers that are covered in the model, and these are covered in our uncertainty analysis.

8. *L170ff: I think this is already discussion*

Following *Nature Communications* style, we have included directly-relevant discussion of results in the Results section.

9. L 178-179: *This sentence seems a bit weird (“improvement or otherwise in...”)*

We have re-written this section (line 194 et seq).

10. L190ff: *this is only limited for GHG, for other aspects (ecosystem services it is not limited to 20 years)!*

We have re-written this section (line 201 et seq).

11. L210: *Why less meat is generally good. It needs to be put in context to the results, where you mention that pasture animals might be an option (I think also the N and P cycling might benefit from some livestock.*

We have added further details of the benefits provided by livestock in terms of nutrient cycling and ecosystem services (lines 211-217).

12. *It is not really clear how N and P limitations have been modelled. There is some legumes, but what about P?*

All the N in our organic systems is provided through biological fixation, manures, atmospheric deposition and any imported feed (lines 257-259), and crop and livestock yields are adjusted according to the resulting available soil N (lines 244-245). We concentrate on N, and not on other plant nutrients, on the basis that N is required in the greatest quantities and its inputs and outputs are the most sensitive to differences between conventional and organic systems. We assume balances of P and K are maintained by applying P and K minerals commonly used in organic systems (lines 259-260) and we calculate the associated GHG impacts of doing this (lines 98-101).

13. *Fig. 1 It should be noted what OSR is. The multiplier for the number scale is a bit odd. Especially it is not clear if milk numbers are related to production or livestock number scale.*

We have added an explanation of OSR and explained that milk impacts are for ‘000 litres (Fig. 1 caption).

14. *Fig 2. N₂O seems to dominate the impacts, while usually it is around 10% of total impacts on a global level - what are the reasons?*

Greater N₂O emissions in the UK compared with globally might be expected as a result of greater fertiliser use per ha and large areas of heavy wet soils. For the conventional baseline, the contributions of individual GHGs to total emissions from crops and livestock are 29, 36 and 34% for N₂O, CH₄ and CO₂, respectively (data in Table 1). According to the FAO, the equivalent global figures are 29, 44 and 27 (Gerber, P.J. et al. 2013. Tackling climate change through livestock – A global assessment of emissions and mitigation opportunities. FAO, Rome).

15. L237 *"not allowed for". Does it mean it is not shown or it is excluded from the analysis? In next figures you show it.*

Emissions due to land use change overseas and enhanced soil carbon sequestration are not allowed for in Fig. 2. However, they are allowed for in Figs 3-4. We have changed the labelling of Fig. 3 to make this clearer.

16. *Fig 3. If you have extensive figure captions, also highlight these numbers only refer to the first 20 years after conversion*

We have added a statement to this effect to the Fig. 3 caption.

17. *Table 1: Spoöit [split] the results also per CH₄, CO₂ and N₂O, to show the total change*

We have added the results for the individual GHGs to Table 1.

18. L281. *At least calculate the total macro nutrients in the two cases.*

We have added a statement that the reduction was similar for both ME and protein (lines 65-66).

19. L331: *Might there be an issue with N2O modelling in the use model?*

No. See responses to Comments 7 and 14 above.

20. L398: *How exactly are the overseas GHG calculated?*

We have revised the section on overseas LUC to make this clearer (lines 383-412).

21. L431: *What about the ripple effect. This means someone else has to cut down the forest and then the organic farmer can transform it from pasture to cropland (and new pastures will be created?) I think this is a big assumption and similar to the assumption that I am not responsible for the GHG emission of the plane, since it is scheduled to fly anyway. At least it should be discussed that indirectly organic agriculture can be responsible for deforestation).*

We believe that this is covered by our new sections and calculations on the Carbon Opportunity Cost of organic conversion (line 179 et seq). We have re-written the discussion of other consequences of organic conversion (lines 201-208).

22. *Whole uncertainty section.*

1- It is not fully clear to me and I expect to have a table with the details in the SI.

2- It seems to be really low uncertainty (+/- 15% in agriculture is really low). Also +/-1% for transport is probably even not covering the same company and transport mode (due to wind etc.)

We have re-written this section (lines 413-442) and added a table to the SI (Table 7). The CoV was 15% for each animal commodity LCA result and 10% for each crop commodity, following Wiltshire et al. (Ref 60). We also point out that the error of 15% is 1 standard deviation divided by the mean. If considering 2 standard deviations, this would of course be $\pm 30\%$. Summing these LCA results for all commodities reduces the overall relative uncertainty as expected. Multiplying them would increase relative uncertainty.

The 1% for transport was a typo: the value used in the simulation was 10%. In any case, it makes a marginal difference to the overall uncertainty, being of low relative magnitude. All transport was assumed to be by road and sea and so affected much less by wind than air travel.

REVIEWER 2

1. *The calculation of the effects on SOC relies solely on the estimates of Gattinger et al. (2012). This is highly problematic, not least due the reasons described by Leifeld et al. (2013; PNAS 110:E984), which criticized the Gattinger paper. Essentially the metaanalyses of Gattinger do not compare equivalent farming systems, as seen from a systems perspective. The organic farms in the metaanalysis had in many cases a higher external carbon import (e.g. of manure and compost) than compared conventional system. In fact, they mostly compared mixed organic systems with mineral fertilizer based conventional systems. Whereas such comparisons may be valuable from a scientific perspective, they cannot be applied to represent the broader agriculture from a systems perspective. I think it is worth quoting Leifeld et al. (2013) on other problems with the metaanalysis that is also relevant to the use of these data in the current context: "In most*

situations, manure would be applied somewhere anyhow, unless used for other purposes. Hence, an increase in soil C in one field (whether in an OF or CF system) is not a net transfer of C from atmosphere to land but a movement of C from one site to another". The current study shows that overall livestock production would decline in the organic farming scenario compared to the conventional one. Therefore, there cannot be an increased amount of manure or compost to be applied on the organic farmed land. This aspect of manure as a mitigation measure was also noted by Powlson (2011; European Journal of Soil Science 62:42–55). The authors needs to revise their calculations taking the issue of the available resources of manure and compost into consideration.

We thank the reviewer pointing out the problems with Gattinger et al.'s C sequestration estimates, which we accept. We have made a new estimate of maximum C sequestration rates following conversion to organic methods using measurements of rates of change in soil C in the National Soil Inventory of England and Wales for rotational grassland, assuming that conversion from continuous arable farming to rotational grass causes comparable C sequestration to organic conversion. We have used Gattinger's lowest sequestration estimate, with no external C inputs, for comparison. We have re-written the sections on C sequestration in the main text (lines 131-159) and Methods (lines 343-382).

- 2. Regarding LUC there is a newly published study by Tim Searchinger (2018, Nature 565:249-253) that in a systematic way addresses the GHG impact of LUC by introducing the concept of opportunity cost. It would be highly pertinent within the current manuscript to make reference to this newly published study and to put the applied method into perspective of the methodology proposed by Searchinger et al. (2018).*

We thank the reviewer for pointing out Seachinger et al. We have calculated Carbon Opportunity Costs (lines 179-192) and re-written the discussion of LUC (lines 194-208).

- 3. Line 128. Change "decreased" to "reduced".*

Done.

- 4. Lines 144-148. These text should be deleted. It does not present results of the analysis. The comments are outside the scope of the study, and in fact the statements can be challenged.*

Deleted.

- 5. Line 192. I would really challenge the concept that application of manure and composts can increase SOC in organic farming. This requires that these materials can be sourced. Figure 1 shows that overall livestock production is lower in the organic farmed scenario than for the conventional scenario. How can there then be more manure applied? The same concern applies for compost?*

We accept that organic farming will not produce vastly more manure or compost. We found livestock numbers increased under organic farming, although production decreased, and we estimate this would have resulting in 12% more manure (lines 136-141), which is a modest amount. A larger potential C sequestration effect is due to the greater use of leys in rotations. We have revised our estimates and discussion of C sequestration as under Comment 1.

- 6. Line 312 Delete "an".*

Done.

- 7. Lines 379-381. I suggest deleting this sentence. I would consider this self-evident*

Done.

REVIEWERS' COMMENTS:

Reviewer #1 (Remarks to the Author):

The paper significantly improved and corrected some mistakes.

I have a few remaining points:

Answer to comment 16 (Fig 3):

In the caption it is: "Note LUC losses and CS gains both only apply over the first few decades following conversion". Not sure this makes clear how it is allocated. In the methods I read "We 393 calculate emissions from the conversion of grassland to crops from the area converted multiplied by

394 LUC emission estimates specified by the British Standards Institute for a range of countries⁵²."

Maybe you can add some additional clarifications. The 4% discount rate only applies to land use, as I understand.

Answer to comment 20 (LUC methods):

It somehow links to the comment above. However, the question is about the land use effect (by the method of Searchinger et al): It would be nice to relate this approach to the standard LCA approach for land use, where basically there is an annual "gap of biodiversity" (or in this case loss of C sequestration) caused by the occupied land. I think, you account for the difference that there is a limitation of sequestration over decades or centuries. Do you need the 4% assumption since you calculate a limitation of the effect (e.g. regrowth of 100 years) and then allocate the lost sequestration over the years using a 4% discount rate?.

I think this needs some more explanation (or if you just use the final factors (as I also read), then also omit the 4% discount rate description).

Reviewer #2 (Remarks to the Author):

The authors have adequately dealt with my comments, and I find that the manuscript reads very well. I only have a few specific comments listed below.

Specific comments

Line 124

It is unclear what the concept "lower estimate" means. Is it a lower reduction in emissions from conversion to organic farming, or is it a lower emission from organic farming compared with conventional.

Line 158

I suggest to change "few years" to "few decades". The change in soil C takes more than years as also calculated on line 380.

Line 184

Change "show" to "show that".

Lines 362-363

Please explain why the sign changes from 0.18 Mg C ha⁻¹ changed to -0.66 Mg CO₂ ha⁻¹ yr⁻¹.

Authors' response to final reviewer comments

We again thank the reviewers and Associate Editor for their comments, all of which we have attended to as follows and as indicated with Track Changes in the MS.

Reviewer #1

1. Answer to comment 16 (Fig 3):

In the caption it is: "Note LUC losses and CS gains both only apply over the first few decades following conversion". Not sure this makes clear how it is allocated. In the methods I read "We 393 calculate emissions from the conversion of grassland to crops from the area converted multiplied by 394 LUC emission estimates specified by the British Standards Institute for a range of countries⁵²." Maybe you can add some additional clarifications. The 4% discount rate only applies to land use, as I understand.

Response

We revised this sentence to 'Note LUC losses and CS gains both only apply over the first few decades following conversion, however a flat rate is applied here'.

2. Answer to comment 20 (LUC methods):

It somehow links to the comment above. However, the question is about the land use effect (by the method of Searchinger et al): It would be nice to relate this approach to the standard LCA approach for land use, where basically there is an annual "gap of biodiversity" (or in this case loss of C sequestration) caused by the occupied land. I think, you account for the inference that there is a limitation of sequestration over decades or centuries. Do you need the 4% assumption since you calculate a limitation of the effect (e.g. regrowth of 100 years) and then allocate the lost sequestration over the years using a 4% discount rate?. I think this needs some more explanation (or if you just use the final factors (as I also read), then also omit the 4% discount rate description).

Response

To avoid confusion, we have deleted 'discounted at 4% per year' from Methods section on COC (new line 406). This point is covered detail by Searchinger et al. along with the full details of the method.

Reviewer #2

1. Line 124

It is unclear what the concept "lower estimate" means. Is it a lower reduction in emissions from conversion to organic farming, or is it a lower emission from organic farming compared with conventional.

Response

We have inserted 'of the effect of conversion to organic farming' (new line 124).

2. Line 158

I suggest to change "few years" to "few decades". The change in soil C takes more than years as also calculated on line 380.

Response

Done (new line 159).

3. Line 184

Change "show" to "show that".

Response

Done (new line 185).

4. Lines 362-363

Please explain why the sign changes from 0.18 Mg C ha⁻¹ changed to -0.66 Mg CO₂ ha⁻¹

yr-1.

Response

The minus sign was erroneous – we have deleted it (new line 364).